# Genetic Modulation of the Erythrocyte Phenotype Associated with Retinopathy of Prematurity—A Multicenter Portuguese Cohort Study

**DOI:** 10.3390/ijms241411817

**Published:** 2023-07-23

**Authors:** Mariza Fevereiro-Martins, Ana Carolina Santos, Carlos Marques-Neves, Hercília Guimarães, Manuel Bicho

**Affiliations:** 1Ecogenetics and Human Health Unit, Environmental Health Institute-ISAMB, Associate Laboratory TERRA, Faculty of Medicine, University of Lisbon, Av. Professor Egas Moniz, 1649-028 Lisboa, Portugal; 2Institute for Scientific Research Bento Rocha Cabral, Calçada Bento da Rocha Cabral 14, 1250-012 Lisboa, Portugal; 3Department of Ophthalmology, Cuf Descobertas Hospital, Rua Mário Botas, 1998-018 Lisboa, Portugal; 4Center for the Study of Vision Sciences, Ophthalmology Clinic, Faculty of Medicine, University of Lisbon, Av. Professor Egas Moniz, Piso 1C, 1649-028 Lisboa, Portugal; 5Department of Gynecology-Obstetrics and Pediatrics, Faculty of Medicine, University of Porto, Alameda Prof. Hernâni Monteiro, 4200-319 Porto, Portugal

**Keywords:** retinopathy of prematurity, preterm infant, gene expression, DNA polymorphism, DNA methylation, histone modifications, epigenetics, biomarker, pathophysiology, retina

## Abstract

The development of retinopathy of prematurity (ROP) may be influenced by anemia or a low fetal/adult hemoglobin ratio. We aimed to analyze the association between *DNA methyltransferase 3 β* (*DNMT3B*) (rs2424913), *methylenetetrahydrofolate reductase* (*MTHFR*) (rs1801133), and *lysine-specific histone demethylase 1A* (*KDM1A*) (rs7548692) polymorphisms, erythrocyte parameters during the first week of life, and ROP. In total, 396 infants (gestational age < 32 weeks or birth weight < 1500 g) were evaluated clinically and hematologically. Genotyping was performed using a MicroChip DNA on a platform employing iPlex MassARRAY^®^. Multivariate regression was performed after determining risk factors for ROP using univariate regression. In the group of infants who developed ROP red blood cell distribution width (RDW), erythroblasts, and mean corpuscular volume (MCV) were higher, while mean hemoglobin and mean corpuscular hemoglobin concentration (MCHC) were lower; higher RDW was associated with *KDM1A* (AA), *MTHFR* (CC and CC + TT), *KDM1A* (AA) + *MTHFR* (CC), and *KDM1A* (AA) + *DNMT3B* (allele C); *KDM1A* (AA) + *MTHFR* (CC) were associated with higher RDW, erythroblasts, MCV, and mean corpuscular hemoglobin (MCH); higher MCV and MCH were also associated with *KDM1A* (AA) + *MTHFR* (CC) + *DNMT3B* (allele C). We concluded that the polymorphisms studied may influence susceptibility to ROP by modulating erythropoiesis and gene expression of the fetal/adult hemoglobin ratio.

## 1. Introduction

Retinopathy of prematurity (ROP) is a vasoproliferative disease of the retina impacting preterm and low-birth-weight infants [1]. Probably due to continued advances in perinatal care that have improved the survival of infants with extreme gestational age (GA), the prevalence of ROP has increased and it remains the second leading cause of childhood blindness, after impaired cortical vision [1,2]. ROP is also associated with other serious ocular complications such as severe ametropias, strabismus, and retinal function disorders [3].

The pathogenesis of ROP is characterized by a first phase of inhibition of retinal vascular growth due to hyperoxia that causes a deficit of hypoxia-inducible factor (HIF)-dependent growth factors, such as vascular endothelial growth factor (VEGF) and erythropoietin. The reduction in IGF-1 and omega 3 polyunsaturated fatty acids, which occurs in this initial phase, contributes to the inhibition of angiogenesis. As the infant grows, increased retinal metabolism in an incompletely vascularized retina leads to the second phase of ROP, characterized by a hypoxia-induced increase in VEGF and erythropoietin, and an increase in IGF-1. This may cause uncontrolled retinal neovascularization, which may lead to retinal detachment [4].

In addition to low GA, low birth weight (BW), and use of supplemental oxygen, which are the well-known risk factors for ROP, genetic and perinatal factors have been associated with its development [5,6].

Several studies show that anemia or low hemoglobin levels in the early life of preterm infants are important risk factors for ROP [7,8]. The treatment of anemia of prematurity, when indicated, remains red blood cell transfusion. However, this may exacerbate oxidative stress in preterm infants with an immature antioxidant system, further increasing the risk of ROP, among other oxidative stress diseases [9,10].

It remains unclear why among preterm infants with similar GA and BW, subject to identical environmental conditions, some develop ROP and others do not. It is known that the individual’s reaction to environmental and internal factors, and, therefore, susceptibility to specific diseases, depends on differences in the genes that control the methylation and demethylation of DNA and histones [11].

Epigenetics comprise several processes that reversibly modulate gene expression without altering the DNA sequence [12].

Methylation reactions are involved in epigenetic regulation, influencing gene expression and protein synthesis [13]. During fetal development and throughout life, DNA methylation, one of the most studied epigenetic mechanisms, regulates numerous critical cellular processes, such as transcriptional regulation, chromosome maintenance, genomic imprinting, and genomic stability [14].

DNA methylation, which is influenced by external and internal factors such as hyperoxia, diet, folate levels, age, and methionine turnover, modifies cellular processes as environmental or internal circumstances change [12,15,16].

The methylation of cytosines followed by guanines in DNA (CpG or CG dinucleotides) is present in the promoters of several genes and regulates canonical gene expression [12]. Promoter demethylation is generally related to gene expression and promoter hypermethylation with gene silencing, although there are exceptions [17].

The DNA methylation process is mediated by a family of enzymes, the DNA methyltransferases (DNMTs), which transfer a methyl group from S-Adenosyl methionine (SAM) to the 5′ position of the DNA cytosine residue in a CpG site [15]. DNMT1, DNMT3A, and DNMT3B are the main DNA methyltransferases [18]. All three DNMTs play a key role in the epigenetic regulation of mammalian retinal development [19].

*DNMT3B* rs2424913 polymorphism is located in the promoter and 3′-untranslated regions of *DNMT3B* gene [18]. This polymorphism regulates the expression of the *DNMT3B* gene, with the minor T allele increasing promoter activity by 30% and affecting the miRNA binding site [14]. Studies report that *DNMT3B* rs2424913 minor T alleles in infants and mothers are significantly associated with preterm birth or a family history of preterm birth [18,20].

Methylenetetrahydrofolate reductase (MTHFR) is a key regulatory enzyme of the folate pathway [14]. The *MTHFR* gene is situated on the short arm of chromosome 1 (1p36.22) [21].

In the study by Nan et al., the TT genotype of the maternal *MTHFR* C677T polymorphism (rs1801133) increased the risk of preterm delivery, whereas the CC genotype of the maternal *MTHFR* A1298C polymorphism (rs1801131) played a protective influence [22].

The enzyme encoded by *MTHFR* generates the SAM radical (a methyl donor) during folate metabolism, while DNMT in the process of DNA methylation uses SAM as a substrate [23].

The DNA methylation profile can be influenced by variations in the concentration or activity of DNMTs and MTHFR, which, in turn, leads to variations in gene expression [23]. Specifically, the T allele of the C677T *MTHFR* (rs1801133) polymorphism is associated with a decrease in the enzymatic activity of MTHFR, and the 149C→T *DNMT3B* (rs2424913) promoter polymorphism is associated with an increased expression of DNMT3B [23].

Another major regulatory epigenetic mechanism is the lysine methylation of the tails of central histones H3 and H4, reversible post-translational alterations of histones linked to transcriptional activation or repression [24,25]. Methyltransferases and demethylases dynamically regulate histone lysine methylation [25]. Methylation of histone lysine residues alters chromatin conformation and controls the expression of genes involved in cell metabolism [26].

The first histone demethylase discovered, the lysine-specific histone demethylase 1A, KDM1A (also known as LSD1, BHC110, and AOF2) is a flavin-dependent monoamine oxidase [25,26]. KDM1A demethylates mono-methylated and di-methylated lysines at histone H3, position 4 (H3K4 me1/2) in promoter regions, activating transcription [24,27]. KDM1A is implicated in repressing transcription through demethylation of H3K9 me1/2 when linked to androgen or estrogen receptors [27,28]. In erythropoiesis, KDM1A was shown to guide erythroid differentiation, mediating the function of the transcription factors GATA-1, TAL1, and C/EBPa [27].

The different epigenetic mechanisms interact in a complex way, contributing to the regulation of gene expression. An example of this is the close relationship between DNA methylation and lysine histone methylation [12].

It is known that the result of a mutation in different individuals may be variable, depending on the interaction with other mutations in different genes. This dependence of the results of mutations on the individual genetic background is called genetic interaction or epistasis. When the genetic interaction of a mutation in a given genetic background leads to a result greater (or better) than expected, there is positive epistasis, whereas when the effect of a mutation is less (or worse) than expected, there is negative epistasis [29]. To the best of our knowledge, there is no literature investigating the possible contribution of genetic polymorphisms related to DNA and histone methylation, and the risk of ROP development.

As the *DNMT3B*, *MTHFR*, and *KDM1A* genes regulate DNA methylation patterns, gene expression, cell proliferation, and erythropoiesis, we hypothesized that polymorphisms in these genes could influence erythropoiesis and be associated with the development of ROP.

Therefore, the aim of our study was to analyze the role of *DNMT3B* (rs2424913), *MTHFR* (rs1801133), and *KDM1A* (rs7548692) polymorphisms in the modulation of erythropoiesis and their association with ROP.

## 2. Results and Discussion

### 2.1. Clinical Characteristics

This study included 396 preterm infants, 238 (60.1%) without ROP and 158 (39.9%) with ROP. Table 1 contains the preterm infants’ principal demographic and clinical characteristics in this study, stratified by the presence or absence of ROP. GA and BW are lower in preterm infants with ROP (*p* < 0.001). The distribution of gender is similar in both groups. The value of Apgar at minute 5 is inferior to 7 in 24 (64.9%) of the preterm infants with ROP (*p* = 0.001). The total days of oxygen ventilation, glycemia ≥ 125, red cell transfusions, and platelets transfusions are statistically different between preterm infants with or without ROP, with preterm infants with ROP experiencing more days of these conditions (*p* < 0.001). Regarding premature diseases, the prevalence of bronchopulmonary dysplasia, necrotizing enterocolitis, periventricular/intraventricular hemorrhage, and patent ductus arteriosus is higher in preterm infants who develop ROP. The mean values of hemoglobin and mean corpuscular hemoglobin concentration are higher in preterm infants without ROP, while red cell distribution width and erythroblasts (×10^9^/L) are lower in this group (Table 2).

### 2.2. Anemia

The presence of anemia in the first week is a risk factor for developing ROP. However, this factor is dependent on the GA (Table 3). The distribution of the polymorphism *DNMT3B* is statistically different regarding the presence or absence of anemia. Preterm infant carriers of allele T or genotype CT have a 1.89- or 1.54-fold risk for the development of anemia, respectively.

### 2.3. DNMT3B, KDM1A, and MTHFR Polymorphism and ROP

The presence of the T allele in *DNMT3B* polymorphism presents a 2.16-fold risk for developing ROP (Table 4). No more significant results are observed when comparing the distribution of different genotypes or alleles and the presence of ROP.

### 2.4. Epistatic Relations and ROP

Epistatic relations between all the studied polymorphisms and ROP are represented in Table 5. However, no significant results are found.

### 2.5. DNMT3B, KDM1A, and MTHFR Polymorphism and Hematological Parameters

The mean value of RDW is statistically different in the three polymorphisms (Table 6). Higher values of RDW are found in carriers of allele C in *DNMT3B*, AA genotype in *KDM1A*, and the CC genotype and CC + TT genotype in *MTHFR*. For erythroblasts, a higher mean value is observed in carriers of the AA genotype for *KDM1A* polymorphism. In all of these cases, the statistical significance continues after the adjustment for GA and days of red cell transfusions.

### 2.6. Epistatic Relations and Hematological Parameters

Table 7 presents the various combinations of epistasis. The mean values of erythroblasts are significantly different in all combinations, with higher values in all the studied polymorphisms. The RDW is higher in the *DNMT3B* (Allele T) + *KDM1A* (AA) and *DNMT3B* (Allele T) + *MTHFR* (CC) + *KDM1A* (AA) compared to other genotypes. The carriers of the allele T of *DNMT3B* and genotype CC of *MTHFR* have lower MCH and MCHC in relation to the other genotypes.

### 2.7. DNMT3B, KDM1A, and MTHFR Polymorphism, Hematological Parameters, and ROP

The association between the studied polymorphisms and the hematological parameters between the two groups (without ROP and with ROP) was also evaluated. In the group without ROP, higher values of MCH and MCV are found in carriers of the allele T of the *MTHFR* polymorphism (Table 8(a)). Contrarily, the non-risk genotypes of *DNMT3B* + *MTHFR* are associated with lower MCV and MCH.

Regarding the group with ROP, higher values of RDW in preterm infants with the genotype: CT (*DNMT3B*), AA (*KDM1A*), and CC + TT (*MTHFR*) are found (Table 9(a)). When evaluating the epistatic association between *MTHFR* and *KDM1A* polymorphisms, higher values of MCV, MCH, RDW, and erythroblasts are found for carriers of the risk genotypes (Table 9(b)). Higher values of erythroblasts are found in carriers of the risk genotype of *DNMT3B* + *MTHFR*. The same hematological parameters are also associated with the risk genotype of *DNMT3B* + *KDM1A* (Table 9(b)). It is observed that preterm infants’ carriers of the three risk genotypes have higher values of MCV, MCH, and erythroblasts (Table 9(b)).

### 2.8. Discussion

Despite recent advances in perinatal care, preterm birth remains associated with ROP, especially in low BW infants [30]. Our study confirms that low GA is associated with ROP among other serious comorbidities, in agreement with the results of several other studies [31,32].

It is well known that low GA, low BW, and the use of supplemental oxygen are the main risk factors for ROP [5]. Regardless of current protocols for managing supplemental oxygen therapy with the aim of preventing hyperoxia, many preterm infants develop severe forms of ROP [33,34].

Anemia or low hemoglobin levels in early life of preterm infants have also been implicated as important risk factors for ROP development [8,35].

At birth, hemoglobin levels are higher than later in life, which occurs to compensate for relative intrauterine arterial hypoxia [36]. A physiological anemia occurs in the first months of life, being exacerbated in very premature infants [36]. This is mainly due to an impaired ability of immature kidneys to increase erythropoietin production, which is exacerbated by frequent phlebotomies for blood collection [37].

Criteria for diagnosing anemia in preterm infants vary throughout the world. In a recent study by Wang et al., hemoglobin values up to 14.5 g/dL in the first week after birth were considered early anemia in preterm infants [38]. Some studies suggest that in very preterm infants, initial hemoglobin levels below 15 g/dL are a risk factor for severe neonatal morbidities [36,39]. In addition, criteria for red blood cell transfusion vary and depend on hemoglobin concentration, clinical status, and days after birth [40,41,42]. In our study, minimum hemoglobin values below 14.5 mg/dL are significantly associated with the development of ROP (odd ratio 3.23), although this association is dependent on GA.

Our study shows significant differences in the blood count of the first week of life between infants who subsequently develop ROP and those who do not. Higher erythroblast counts and RDW, as well as lower mean hemoglobin levels and MCHC, are significantly associated with the development of ROP, which is in line with the work we previously reported [43]. Of these parameters related with anemia, only mean hemoglobin levels lose statistical significance after adjustment for GA and number of days of red blood cell transfusion, risk factors associated with ROP after logistic regression analysis. In the study by Lundgren et al., although low mean hemoglobin levels in the first week of life were significantly associated with ROP, this outcome was also dependent on GA [35]. However, we emphasize that in the same study, the number of days with anemia in the first week of life and the number of transfusions were independent risk factors for ROP requiring treatment.

Several other studies associated blood transfusions and anemia in early life with ROP and ROP requiring treatment [44,45]. In our study, the total number of days of red blood cell transfusion is significantly associated with the development of ROP. Newborn’s hemoglobin is predominantly fetal hemoglobin (HbF), with a higher affinity for oxygen than adult hemoglobin [8]. Transfusions are performed with adult blood containing almost 100% hemoglobin A (HbA), leading to a replacement of the physiological content of HbF by HbA, and a shift to the right in the oxygen–hemoglobin dissociation curve [46]. A higher proportion of adult hemoglobin may lead to increased oxygen bioavailability, causing relative tissue hyperoxia [8]. Stuchfield et al. found significantly lower levels of HbF in preterm infants with ROP [47]. In addition, HbF prevents the release of heme groups, has a greater pseudo peroxidase activity with quicker reconversion of reactive ferryl-heme, and even greater capability to produce unbound nitric oxide through oxidative denitrosylation [34]. The PacIFiHER study shows that low levels of HbF in preterm infants increases the risk for ROP and is correlated with poorer systemic oxygenation indices [48].

The one-carbon cycle comprises interconnected biochemical pathways, including the folate and the methionine pathways (Figure 1). These biochemical pathways produce methyl groups involved in epigenetic regulation and several crucial processes, including DNA synthesis, amino acid homeostasis, and antioxidant production [13].

Methylation processes play a key role in gene expression, genomic stability and development [49]. Variation in one-carbon metabolism by genetic factors, namely, MTFHR polymorphism, may influence infant methylation [50].

Premature birth has been associated with genetic factors, such as the *DNMT3B* polymorphism (rs2424913) in infants and mothers, and the *MTHFR* polymorphism (rs1801133) in mothers [18,20,22].

Epigenetic factors drive the differentiation of red blood cells. During erythroid maturation, changes in histones have been implicated in the regulation of nuclear condensation and gene expression [51,52]. KDM1A, a lysine-specific histone demethylase, plays a key role in erythroid maturation, erythroid gene regulation, and HbF silencing [52,53].

Our aim was to analyze the influence of genes associated with premature birth, cell differentiation, and proliferation, and erythropoiesis in the modulation of erythrocyte parameters associated with ROP (intermediate phenotype) and in the development of ROP (distant phenotype). For this purpose, we studied polymorphisms of the *DNMT3B* (rs2424913), *MTHFR* (rs1801133), and *KDM1A* (rs7548692) genes, involved in major epigenetic regulatory mechanisms.

When we analyzed the distribution of the three genetic polymorphisms studied among the two groups of infants, with and without ROP, the only significant association found is between the presence of the minor allele (T allele) of the *DNMT3B* rs2424913 polymorphism and the development of ROP, but this result is dependent on GA (Table 4). This polymorphism results in an increased expression of DNMT3B, influencing the DNA methylation profile [23]. However, as this polymorphism is associated with preterm delivery, the link with ROP is controversial, and may depend on the prematurity itself and not represent a direct risk factor for ROP [18,20]. Even so, we cannot exclude the existence of possible pathophysiological mechanisms associated with DNA hypermethylation, related to the T allele of the *DNMT3B* rs2424913 polymorphism, which may influence the susceptibility to both premature birth and the development of ROP.

It should also be noted that the T allele of the *DNMT3B* rs2424913 polymorphism is significantly associated with anemia (Table 3). However, when adjusted for GA, the association of this polymorphism with anemia loses significance, being just a trend.

We find positive epistasis between *DNMT3B* (T allele) and *MTHFR* genotype (CC). In the study population, carriers of these polymorphisms have a decrease in MCHC and MCH (Table 7). Particularly in the group that does not develop ROP, in carriers of the same polymorphisms, a decrease in MCH and MCV is observed (Table 9(a)). These results suggest that this epistatic relationship does not influence the development of ROP. It is possible that this association may result from reduced hemoglobin synthesis due to lower gene expression resulting from increased DNA methylation caused by both polymorphisms.

In the study population, the C allele of the *DNMT3B* polymorphism is significantly and independently associated with higher RDW (Table 6). This may be due to the lower activity of DNMT3B related to the presence of the C allele, with less DNA methylation and, consequently, higher gene transcription and expression. Data from human umbilical-cord-derived erythroid cells and mouse fetal liver erythroblasts show that evolution from erythroid progenitors to matured erythroblasts is correlated with significant demethylation [54]. Both DNMT3B and DNMT3A are significantly decreased during the differentiation phase [54]. Although the association of increased RDW with the C allele of *DNMT3B* is not significant in the group that develop ROP, increased RDW is significantly and independently associated with the development of ROP when the C allele of *DNMT3B* is associated with the AA genotype of *KDM1A* (Table 8(b) and Table 9(b)).

Extensive research shows that KDM1A is essential for stem cell differentiation and maintenance, embryonic development, and erythropoiesis [28,55]. A reduced level of KDM1A in human embryonic stem cells is associated with impairment of cell cycle progression and abnormal expression of developmentally controlled genes [28].

KDM1A plays a critical role in the regulation of erythroid gene and erythroid maturation [52]. *KDM1A* knockdown in mice or deletion of *KDM1A* in hematopoietic stem cells results in anemia and reduction in mature blood cells [53]. Inhibition or knockdown of *KDM1A* in human erythrocyte precursors induces high levels of HbF, as KDM1A is a member of the γ-globin repressor complexes [56]. In this regard, it is known that a moderate increase in HbF levels, for example in patients with sickle cell disease or β-thalassemia, can modify the clinical course of the disease and, therefore, KDM1A represents a target in the treatment of β- globinopathies [34,53,56].

The AA genotype of *KDM1A* rs7548692 polymorphism is significantly associated with increased RDW and erythroblasts, parameters associated with ROP, in the study population (Table 6). Furthermore, we found that the association between this genotype and increased RDW remains significant and independent only for preterm infants who later develop ROP (Table 8(b)).

Unsal et al. and Çömez et al. found a significant association between the increase in RDW and the development of ROP in the first two weeks and in the fourth week of life, respectively [57,58]. RDW represents the coefficient of variation of red blood cell volume, being a reliable index of anisocytosis [57,59]. It has been suggested that the ineffective maturation of red blood cells can disrupt their membrane, leading to increased RDW. Also, the inflammatory process, mediated by cells and cytokines, can impair bone marrow function, leading to the release of premature erythrocytes into the circulation, with a consequent increase in RDW [60]. RDW can additionally be increased due to red blood cell transfusion [59]. Besides its traditional clinical use to assess different types of anemia, RDW has been associated with the presence and worse prognosis of a wide range of human pathologies, including in infants [59,61]. RDW has been proposed as an important biomarker of erythrocyte damage, influencing the prognosis of critical illnesses, possibly by affecting tissue oxygen supply [61].

MTHFR is a main regulatory enzyme of the one-carbon metabolic pathway. The one-carbon metabolic pathway includes the folate and methionine pathways, which provide methyl groups for both methylation and nucleotide synthesis. The *MTHFR* rs1801133 polymorphism affects the function of the MTHFR enzyme. The TT and CT genotypes of this polymorphism correlate with elevated plasma homocysteine levels due to a decrease in MTHFR enzymatic activity of about 70% and 40%, respectively, compared to the wild-type CC genotype [14,62].

Another relevant finding of our study is that for carriers of both the AA genotype of *KDM1A* and the CC genotype of *MTHFR*, there is a significant increase in RDW and erythroblasts and that these associations only remain significant and independent in the ROP group (Table 7 and Table 9(b)).

The increase in both RDW and erythroblasts or other erythrocyte precursors may result from increased erythropoietic activity, with the release of immature erythrocytes from the bone marrow, usually in an attempt to compensate for anemia. Niranjan et al. and Lubetsky et al. found in their studies that an increase in the nucleated red blood cell count on the first day of life was significantly associated with the later development of ROP [63,64]. The CC genotype of *MTHFR* has a high activity of the MTHFR enzyme that leads to an activation of the folate cycle, important for the synthesis of nucleic acids and, as such, for cell proliferation. Furthermore, both KDM1A and MTHFR are flavin adenine dinucleotide (FAD)-dependent. Oxidative stress leads to increased oxidized glutathione (GSSG), and FAD is required for its recycling (GSH) through glutathione reductase activity. Methionine adenosyl transferase (MAT), the enzyme that converts methionine to SAM, is also controlled by the reduced glutathione/ oxidized glutathione (GSH/GSSG) ratio [65]. As the *MTHFR* CC and *KDM1A* AA genotypes are those with the highest enzymatic activity, they should also be the least affected by the lower availability of FAD associated with oxidative stress. We hypothesize that, at least in part, these are the reasons for a synergistic effect of the association of the AA genotype of *KDM1A* and the CC genotype of *MTHFR* regarding RDW and erythroblast levels in preterm infants who develop ROP.

We find that the association of increased RDW with the *MTHFR* rs1801133 polymorphism in the study population that develops ROP is valid for homozygotes (CC + TT), which have opposite levels of MTHFR enzyme activity (Table 6 and Table 8(b)). As we mentioned before, the CC genotype of *MTHFR* rs1801133 polymorphism has an increased enzyme activity and may, therefore, be associated with an increase in cell proliferation. In contrast to the CC genotype, the TT genotype of *MTHFR* (rs1801133) has less enzyme activity, thereby producing less methionine, which is required for the synthesis of SAM, the universal primary donor of methyl groups for methylation. This result suggests that in carriers of the TT genotype, lower DNA methylation in erythropoiesis regulatory genes may facilitate the activation of erythropoiesis in response to anemia, with a consequent increase in anisocytosis and, therefore, in RDW.

In carriers of the three risk genotypes, C allele of *DNMT3B*, genotype AA of *KDM1A*, and genotype CC of *MTHFR*, we observe that the increase in MCV, MCH, and erythroblasts is significantly associated with the development of ROP, although the association with erythroblasts is not independent (Table 9(b)). The interaction between these epigenetic mechanisms is complex and influences gene expression.

MCV defines the size of red blood cells while MCH reveals the amount of hemoglobin per red blood cell. Increased MCV is also found in megaloblastic anemia secondary to folic acid (vitamin B9) or cobalamin (vitamin B12) deficiency, in addition to anemia associated with chronic pathologies and less frequent hereditary diseases. Increased MCV may result from an acute erythropoietic stimulus that leads to recruitment of erythroid progenitors rather than relatively differentiated erythroids or that accelerates the erythropoietin-induced hemoglobin synthesis, causing nuclear degeneration of erythrocytes [66,67].

Elevated levels of MCH and MCV can be used as markers of macrocytosis. In macrocytosis, the imbalance between the nucleus and the cytoplasm can make red blood cells less flexible, hindering their flow in the microcirculation and impairing the important antioxidant function of the erythrocyte membrane [67].

In summary, the studied genetic polymorphisms are not directly associated with the development of ROP. However, to the best of our knowledge, our results suggest for the first time a relationship between the genetic polymorphisms of the *DNMT3B* (rs2424913), *MTHFR* (rs1801133), and *KDM1A* (rs7548692) genes and the development of ROP indirectly.

The increase in RDW is associated with the development of ROP for both the AA genotype of *KDM1A* and for the *MTHFR* homozygotes. The increase in RDW and erythroblasts is associated with the development of ROP for carriers of the AA genotype of *KDM1A* and CC genotype of *MTHFR*. The AA genotype of *KDM1A*, alone or associated with the CC genotype of the *MTHFR* or the C allele of *DNMT3B*, is associated with increased RDW in preterm infants with ROP.

Our results highlight a central role of the AA genotype of *KDM1A* in the hematological phenotype (intermediate phenotype) associated with ROP. It is well-known that KDM1A plays a crucial role in regulating erythropoiesis and inhibiting HbF synthesis. Several studies show that in addition to anemia, the reduction in the HbF/HbA ratio in the early life of preterm infants is an important risk factor for the development of ROP. Based on these assumptions, we admit that the influence of the genotype AA of *KDM1A* on the hematological phenotype associated with ROP that we found may be related to a predisposition of this genotype to anemia and possibly also to a reduction in HbF. HbF has a higher affinity for oxygen than HbA and may also have a protective role against oxidative stress. Therefore, reduced HbF may result from repeated transfusions with adult blood but is possibly aggravated by the influence of genetic factors.

As for limitations of this study, the main one is that the HbF measurement is not performed, which would confirm our conclusions. In addition, as this is an observational study, specific moments were not defined for the study of erythrocyte parameters during the first week of life. Another limitation of our study is its relatively small size, as associations of polymorphisms with certain conditions normally require studies on a larger scale. It is possible that associations between the studied polymorphisms and the development of ROP will be revealed in a larger sample.

Although the associations found between the genetic polymorphisms studied and the erythrocyte parameters associated with ROP are statistically significant, further studies are needed to confirm the results obtained. Our work may be a starting point for more comprehensive studies, also involving fetal hemoglobin and erythropoietin measurements, which may help clarify the genetic susceptibility to ROP.

We emphasize that, if our results are confirmed, approaches that may help to prevent a significant drop in HbF in very preterm infants may reduce the risk of ROP.

The findings of our study may help to clarify the pathophysiological mechanisms associated with ROP. Furthermore, they may contribute to the discovery of genetic and circulating biomarkers that identify earlier premature infants with a higher risk of developing ROP. Thus, they may also have their applicability in the development of more appropriate screening and treatment strategies for this pathology, according to individual genetic susceptibility.

## 3. Materials and Methods

### 3.1. Population

The current study is an observational, prospective, and multicenter study, involving 8 neonatal intensive care units from the following Portuguese hospital centers: Centro Hospitalar Universitário de Lisboa Norte, Lisbon; Hospital Prof. Doutor Fernando Fonseca, Amadora; Centro Hospitalar Universitário de São João, Porto; Centro Materno Infantil do Norte belonging to the Centro Hospitalar Universitário do Porto, Porto; Hospital de Braga, Braga; Hospital da Senhora da Oliveira, Guimarães; Maternidade Bissaya Barreto, and Maternidade Daniel de Matos belonging to the Centro Hospitalar Universitário de Coimbra, Coimbra.

Patients screening and enrollment occurred between 19 of November of 2018 and 21 of July of 2021 in the above-mentioned hospitals. The inclusion criteria were preterm infants (1) born before 32 weeks of GA and/or (2) born with BW of less than 1500 g, regardless of sex and race. Preterm infants with (1) major congenital malformations, (2) ophthalmological pathologies, congenital or acquired (during the first 12 weeks of life) not related to ROP (except for conjunctivitis, keratitis, and congenital nasolacrimal duct obstruction), (3) death before the first ROP screening, (4) insufficient clinical data due to patient transfer to another hospital, and (5) absence of informed consent from parents or legal guardians, were excluded from the study.

This cohort included 396 preterm infants that were eligible according to the inclusion and exclusion criteria of the study, from which 198 (50.0%) were females and 158 (39.9%) were diagnosed with ROP.

### 3.2. ROP Screening and Ophthalmological Data Collection

Preterm infants were screened for ROP and other ophthalmological pathologies at 31 to 33 weeks of post-menstrual age (PMA) or 4 to 6 weeks after birth by qualified ophthalmologists at the neonatal intensive care unit. The screening and diagnosis followed the guidelines of the latest consensus on ROP of the Portuguese Society of Neonatology [68] and International ROP Classification (ICROP), published in 1984 [69], and revisited in 2005 [70], mainly: (1) adequate mydriasis by the administration of eye drops; (2) indirect binocular ophthalmoscopy or digital fundus retinography (RetCam) or both; (3) the results of each test obtained for each eye separately documented and described in detail, including the location, the severity of retinopathy (stages), the extent of lesions in the active phase, and the presence or absence of plus-disease; (4) posterior evaluation and follow-up examinations were carried on according to the existence and severity of ROP and were repeated until complete retinal vascularization or until complete remission of ROP after treatment. Preterm infants with ROP type 1 diagnosis were treated with LASER photocoagulation or anti-VEGF, depending on the location of the lesion in the retina and the hospital center, according to the guidelines introduced by the Early Treatment for Retinopathy of Prematurity study (ETROP) [71].

### 3.3. Genetic Polymorphism Identification

The genetic study of infants was carried out from a buccal swab, or when this was not enough, using the surplus of blood collected in the first four weeks of life after carrying out routine laboratory parameters. Genotyping was performed using a MicroChip DNA on a high-throughput platform using iPlex MassARRAY^®^ technology from Agena Bioscience (San Diego, CA, USA). The PCR reaction was specific for each allelic variant. The obtained genotypes were read by MALDI-TOF mass spectrometry. Genotyping data were analyzed by HeartGenetics’ EARTDECODE^®^ software system. The allele and genotype frequencies, and the balance of their proportions were analyzed according to Hardy and Weinberg.

### 3.4. Demographic and Clinical Data

Clinical, laboratory, and demographic data were collected from the clinical process. The blood count was determined using standardized methods. Blood count included: hemoglobin (g/dL); red blood cell distribution width (RDW) (%); erythroblasts (×10^9^/L); mean corpuscular hemoglobin (MCH) (pg); mean corpuscular hemoglobin concentration (MCHC) (pg); mean corpuscular volume (MCV) (fl).

Clinical data included: (1) gestation data: assisted reproductive technologies, twin or multiple births; (2) birth data: GA (weeks), BW (g), type of delivery, gender, Apgar at minute 5; (3) prenatal data: prenatal steroids, prenatal magnesium, maternal chorioamnionitis; (4) days of oxygen ventilation; (5) days with glycemia ≥ 125 mg/dL; (6) days of red blood cell and platelets transfusions; (7) comorbidities such as bronchopulmonary dysplasia, necrotizing enterocolitis, periventricular–intraventricular hemorrhage (PIVH), patent ductus arteriosus with hemodynamic significance, periventricular leukomalacia.

### 3.5. Statistical Analysis

The normality of variables was tested using the Kolmogorov–Smirnov test, and the values were presented using the median and interquartile range. The Pearson χ^2^ test was used to evaluate the significant differences between groups. To compare groups, the Mann–Whitney test was used. Logistic regression analysis and the corresponding 95% CI were calculated using a binary dependent variable to model the probability of a risk factor for No ROP/ROP and genetic polymorphism. A multivariate regression (backward conditional) was performed using variables that were significant in univariate regression and clinically meaningful. All the results were adjusted for these variables (GA and days of red blood cell transfusions). Statistical analysis was performed with the SPSS program with a significant value of *p* < 0.05.

### 3.6. Ethics Approval

The current study was performed in accordance with the ethical standards as laid down in the 1964 Declaration of Helsinki and its later amendments or comparable ethical standards. The study, protocol, data collection form, and informed consent were approved by the Scientific Council of the Faculty of Medicine of the University of Lisbon and by the Ethics Committees of participating hospital centers (Centro Hospitalar Universitário de Lisboa Norte and Centro Académico de Medicina de Lisboa (CAML), protocol code 340/2018, approval on 19 October 2018; Ethics Committee of Hospital da Senhora da Oliveira Guimarães, protocol code 01/19-CAc, approval on 11 January 2019; Ethics Committee of Centro Hospitalar Universitário de São João, protocol code 20-19, approval on 21 January 2019; Ethics Committee of Hospital Professor Doutor Fernando Fonseca, protocol code 70/2018, approval on 12 December 2018; Ethics Committee of Hospital de Braga, protocol code 15/2019, approval on 20 March 2019; Ethics Committee of Centro Hospitalar e Universitário de Coimbra, protocol code CHUC-014-019, approval on 16 April 2019; Ethics Committee of Centro Hospitalar Universitário do Porto, protocol code 031-DEFI/032-CE, approval on 21 May 2019). Collected data were stored and treated as confidential clinical information under the anonymity of the participants. Written informed consent was obtained from the parents of all infants included in the study.

## Figures and Tables

**Figure 1 ijms-24-11817-f001:**
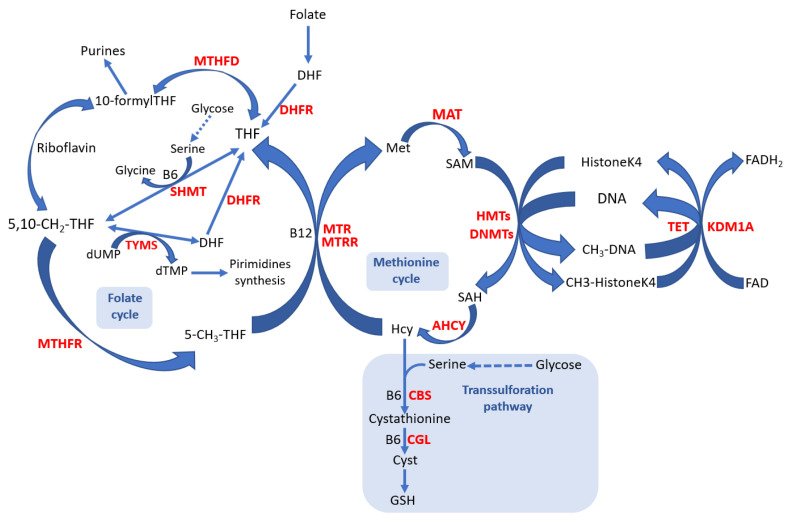
Overview of the one-carbon metabolic pathways. The one-carbon metabolism includes the folate and methionine cycles. Dietary folate gives rise to THF, a one-carbon (methyl groups) acceptor. THF accepts one-carbon units from amino acids such as serine. The resulting methylated THF provides one-carbon units for important cellular functions, such as nucleic acid synthesis and methylation reactions through the methionine recycling pathway. Enzymes (in red bold): DHFR, dihydrofolate reductase; MTHFD, methylenetetrahydrofolate dehydrogenase; MTHFR, methylenetetrahydrofolate reductase; MTR, methionine synthase; MTRR, methionine synthase reductase; SHMT, serine hydroxyl methyltransferase; TYMS, thymidylate synthase; MAT, methionine adenosyl transferase; HMTs, histone methyltransferases; DNMTs, DNA methyltransferases; AHCY, S-adenosyl-L-homocysteine hydrolase; TET, ten-eleven translocation enzymes; KDM1A, lysine-specific histone demethylase 1A; CBS, cystathionine beta-synthase; CGL, cystathionine gamma-lyase. Metabolites: DHF, dihydrofolate; THF, tetrahydrofolate; 10-formylTHF, 10-formyl tetrahydrofolate; 5,10-CH2-THF, 5,10-methylenetetrahydrofolate; 5-CH3-THF, 5-methyltetrahydrofolate; Met, methionine; SAM, S-adenosylmethionine; SAH, S-adenosylhomocysteine; Hcy, homocysteine; dUMP, deoxyuridine monophosphate; dTMP, deoxythymidine monophosphate; Cyst, cysteine; GSH, glutathione. Cofactors: B6, vitamin B6; B12, vitamin B12; riboflavin, vitamin B2.

**Table 1 ijms-24-11817-t001:** Demographic and clinical characteristics stratified by the presence or absence of ROP.

Clinical Characteristics n (%) or n; Median, Interquartile Range	Total	No ROP	ROP	*p*-Value	*p*-Value *
Number of individuals n (%)	396	238 (60.1)	158 (39.9)	n.a.	n.a.
Gestational age (weeks)	396; 29.5, 3.1	238; 30.4, 2.3	158; 28.0, 2.9	**<0.001**	n.a.
Birth weight (g)	396; 1187.5, 444.3	238; 1300.0, 381.0	158; 980.0, 434.3	**<0.001**	**0.016**
Gender					
Male	198 (50.0)	126 (52.9)	72 (45.6)	0.182	0.139
Female	198 (50.0)	112 (47.1)	86 (54.4)		
Apgar at minute 5					
≥7	353 (90.5)	223 (94.5)	130 (84.4)	**0.001**	0.298
<7	37 (9.5)	13 (5.5)	24 (15,6)		
Assisted reproductive technologies (yes)	48 (14.3)	26 (54.2)	22 (45.8)	0.753	0.724
Twin or multiple births (yes)	112 (28.4)	77 (68.8)	35 (31.3)	**0.04**	0.240
Type of delivery (Cesarean)	269 (68.3)	165 (61.3)	104 (38.7)	0.538	0.705
Maternal chorioamnionitis (yes)	43 (11.0)	23 (53.5)	20 (46.5)	0.328	0.999
Prenatal steroids (yes)	366 (92.7)	223 (60.9)	143 (39.1)	0.332	0.169
Prenatal magnesium (yes)	147 (63.4)	81 (55.1)	66 (44.9)	0.168	0.322
Days of oxygen ventilation	394; 16.5, 43	236; 7.0, 24	158; 42.5, 51	**<0.001**	0.438
Days with glycemia ≥125 mg/dL	396; 1.0, 3	238, 1.0, 2	158; 2.5, 5	**<0.001**	0.400
Days of red cell transfusions	396; 0.0, 1	238; 0.0, 0	158; 1.0, 4	**<0.001**	n.a.
Days of platelets transfusions	396; 0.2, 0	238; 0.1, 0	158; 0.4, 0	**<0.001**	0.082
Bronchopulmonary dysplasia (yes)	81 (20.7)	22 (27.2)	59 (72.8)	**<0.001**	0.978
Necrotizing enterocolitis (yes)	23 (5.9)	9 (39.1)	14 (60.9)	**0.045**	0.847
Periventricular/intraventricular hemorrhage (yes)	47 (12.1)	16 (34.0)	31 (66.0)	**<0.001**	0.478
Patent ductus arteriosus (yes)	55 (14.1)	15 (27.3)	40 (72.7)	**<0.001**	0.218
Periventricular leukomalacia (yes)	12 (3.1)	4 (33.3)	8 (66.7)	0.071	0.656

n.a., not applicable; *p*-value *, *p*-value adjusted for gestational age and days of red cell transfusions.

**Table 2 ijms-24-11817-t002:** Hematological parameters stratified by the presence or absence of ROP.

Hematological Parameters n (%) or n; Median, Interquartile Range	Total	No ROP	ROP	*p*-Value	*p*-Value *
Hemoglobin (g/dL)	343	197; 16.1, 3.1	146; 14.8, 3.4	**<0.001**	0.240
Red cell distribution width (%)	325	184; 16.5, 2.2	141; 17.1, 2.9	**0.001**	**0.031**
Erythroblasts (× 10^9^/L)	310	176; 7.8, 15.9	134; 14.8, 32.2	**<0.001**	**0.014**
Mean corpuscular hemoglobin (pg)	326	185; 37.7, 3.3	141; 37.3, 4.0	0.146	0.579
Mean corpuscular hemoglobin concentration (pg)	328	186; 35.2, 1.6	142; 33.9, 2.7	**<0.001**	**<0.001**
Mean corpuscular volume (fl)	324	184; 107.4, 9.6	140; 108.6, 12.2	**0.022**	**0.026**

*p*-value *, *p*-value adjusted for gestational age and days of red cell transfusions.

**Table 3 ijms-24-11817-t003:** (a) Prevalence of anemia ^1^ in preterm infants regarding ROP development; (b) prevalence of anemia ^1^ in preterm infants regarding the genotype of *DNMT3B*.

**(a)**						
	**Without ROP** **N (%)**	**With ROP** **N (%)**	**OR [CI 95%]**	** *p* ** **-Value**	** *p* ** **-Value ***	** *p* ** **-Value #**
Hemoglobin ≥ 14.5 g/dL	148 (63.8)	55 (35.0)	1	**<0.001**	0.157	0.063
Hemoglobin < 14.5 g/dL	84 (36.2)	102 (65.0)	**3.23 [2.14–4.99]**			
**(b)**						
*DNMT3B* N (%)	Hemoglobin ≥ 14.5 g/dL	Hemoglobin < 14.5 g/dL	OR [CI 95%]	*p*-value	*p*-value *	*p*-value #
CC	65 (33.0)	38 (20.7)	1	**0.008**	0.073	**0.043**
Allele T	132 (67.0)	146 (79.3)	**1.89 [1.19–3.01]**			
TT	46 (23.4)	46 (25.0)	n.a	0.721	0.696	0.664
Allele C	151 (76.6)	138 (75.0)	n.a			
CC + TT	111 (56.3)	84 (45.7)	1	**0.041**	0.183	0.144
CT	86 (43.7)	100 (54.3)	**1.54 [1.03–2.30]**			

^1^ Anemia defined as hemoglobin below 14.5 g/dL in the first week of life; n.a., not applicable; *p*-value *, *p*-value adjusted for gestational age; *p*-value #, adjusted for days of red cell transfusions.

**Table 4 ijms-24-11817-t004:** Distribution of the genetic polymorphisms (*DNMT3B*, *KDM1A*, and *MTHFR*) in the study population according to the development of ROP (with or without ROP).

Genetic Polymorphisms						
Polymorphism	Genotype	Without ROP N (%)	With ROP N (%)	OR [CI 95%]	*p*-Value	*p*-Value *
*DNMT3B*	TT	52 (22.2)	41 (26.6)	n.a.	0.333	0.412
	Allele C	182 (77.8)	113 (73.4)	n.a.		
	CC	74 (31.6)	33 (21.4)	1	**0.029**	0.193
	Allele T	160 (68.4)	121 (78.6)	**2.16 [1.06–2.72]**		
	CT	108 (46.2)	80 (51.9)	n.a.	0.299	0.695
	CC + TT	126 (53.8)	74 (48.1)	n.a.		
*KDM1A*	AA	32 (13.5)	24 (15.2)	n.a.	0.661	0.406
	Allele T	205 (86.5)	134 (84.8)	n.a.		
	TT	98 (41.4)	65 (41.1)	n.a.	0.967	0.393
	Allele A	139 (58.6)	93 (58.9)	n.a.		
	TA	107 (45.1)	69 (43.7)	n.a.	0.836	0.151
	AA + TT	130 (54.9)	89 (56.3)	n.a.		
*MTHFR*	TT	18 (7.7)	14 (8.9)	n.a.	0.709	0.679
	Allele C	216 (92.3)	143 (91.1)	n.a.		
	CC	128 (54.7)	73 (46.5)	n.a.	0.122	0.554
	Allele T	106 (45.3)	84 (53.5)	n.a.		
	CT	88 (37.6)	70 (44.6)	n.a.	0.173	0.700
	CC + TT	146 (62.4)	87 (55.4)	n.a.		

*DNMT3B, DNA methyltransferase 3 β; KDM1A, lysine-specific histone demethylase 1A; MTHFR, methylenetetrahydrofolate reductase*; n.a., not applicable; OR, odds ratio; *p*-value *, *p*-value adjusted for gestational age and days of red cell transfusions.

**Table 5 ijms-24-11817-t005:** Epistatic relations between all polymorphisms and ROP in the study population.

Epistatic Relations between Polymorphisms and ROP					
Epistatic Relations	No ROP N (%)	ROP N (%)	OR [CI 95%]	*p*-Value	*p*-Value *
*DNMT3B* (Allele T) + *MTHFR* (CC) + *KDM1A* (AA)	10 (4.3)	13 (8.5)	n.a.	0.123	n.a.
Other genotypes	221 (95.7)	140 (91.5)	n.a.		
*DNMT3B* (Allele T) + *KDM1A* (AA)	19 (8.1)	19 (12.4)	n.a.	0.168	n.a.
Other genotypes	215 (91.9)	134 (87.6)	n.a.		
*DNMT3B* (Allele T) + *MTHFR* (CC)	88 (37.9)	61 (39.6)	n.a.	0.750	n.a.
Other genotypes	144 (62.1)	93 (60.4)	n.a.		n.a.
*KDM1A* (AA) + *MTHFR* (CC)	18 (7.7)	15 (9.6)	n.a.	0.579	n.a.
Other genotypes	215 (92.3)	142 (90.4)	n.a.		n.a.

*DNMT3B, DNA methyltransferase 3 β; KDM1A, lysine-specific histone demethylase 1A; MTHFR, methylenetetrahydrofolate reductase*; n.a., not applicable; OR, odds ratio. *p*-value *, *p*-value adjusted for gestational age and days of red cell transfusions.

**Table 6 ijms-24-11817-t006:** Genetic polymorphisms (*DNMT3B*, *KDM1A*, and *MTHFR*) and erythrocyte parameters of the study population. Influence of the genetic polymorphisms on the erythrocyte parameters.

Polymorphism	Genotype	RDW (%) n; Median, Interquartile Range	*p*-Value	*p*-Value *	Erythroblasts (× 10^9^/L) n; Median, Interquartile Range	*p*-Value	*p*-Value *	MCH (pg) n; Median, Interquartile Range	*p*-Value	*p*-Value *	MCHC (pg) n; Median, Interquartile Range	*p*-Value	*p*-Value *
*DNMT3B*	TT	80; 16.0, 1.6	**0.036**	**0.025**	75; 10.1, 19.4	0.867	0.307	80; 37.1, 4.6	0.274	0.232	83; 34.8, 2.3	0.895	0.965
	Allele C	237; 16.7, 2.3			228; 10.8, 22.8			238; 37.5, 3.3			237; 34.9, 2.4		
	CC	86; 16.4, 2.2	0.592	0.414	82; 10.6, 15.7	0.221	0.564	86; 37.4, 2.8	0.279	0.291	86; 35.2, 2.3	**0.044**	0.153
	Allele T	231; 16.1, 1.9			221; 10.4, 27.1			232; 37.5, 4.0			234; 34.8, 2.3		
	CT	151; 17.0, 2.9	0.179	0.196	146; 11.0, 31.6	0.218	0.665	166; 37.5, 3.9	0.989	0.912	169; 34.8, 2.3	0.095	0.216
	CC + TT	166; 16.6, 2.1			157; 10.2, 16.6			152; 37.3, 3.6			151; 35.1, 2.2		
*KDM1A*	AA	41; 17.1, 2.8	**0.037**	**0.039**	40; 18.5, 34.9	**0.019**	**0.044**	41; 38.2, 4.4	0.569	0.583	42; 34.8, 2.1	0.728	0.624
	Allele T	283; 16.2, 1.9			269; 9.4, 34.9			28437.5, 3.4			285; 34.9, 2.4		
	TT	135; 16.4, 1.8	0.517	0.454	130; 9.5, 24.5	0.772	0.555	135; 37.6, 3.7	0.581	0.430	137; 34.9, 2.5	0.845	0.409
	Allele A	189; 16.2, 2.2			179; 10.9, 22.0			190; 37.4, 3.7			190; 34.9, 2.2		
	TA	148; 16.6, 2.2	0.452	0.510	139; 9.4, 17.3	0.193	0.057	149; 37.3, 3.3	0.206	0.253	148; 34.9, 2.3	0.194	0.627
	AA + TT	176; 16.8, 2.6			170; 12.6, 28.9			176; 37.6, 3.9			179; 34.9, 2.5		
*MTHFR*	TT	25; 15.8, 1.2	0.154	0.103	24; 5.6, 17.9	0.131	0.197	25; 38.1, 4.7	0.473	0.535	25; 34.9, 2.0	0.565	0.414
	Allele C	295; 16.3, 2.1			282; 11.2, 23.1			296; 37.4, 3.7			298; 34.9, 2.4		
	CC	162; 16.6, 2.6	**0.004**	**0.002**	157; 23.5, 27.8	**0.004**	**0.090**	162; 37.3, 3.4	0.115	0.095	165; 34.8, 2.5	0.132	0.124
	Allele T	158; 16.0, 2.0			149; 8.2, 17.3			159; 37.7, 3.8			158; 34.9, 2.0		
	CT	133; 16.6, 2.3	**0.031**	**0.023**	125; 8.3, 18.1	**0.038**	0.295	134; 37.6, 3.6	0.227	0.173	133; 34.9, 2.0	0.224	0.064
	CC + TT	187; 16.8, 2.7			181; 13.4, 26.7			187; 37.4, 3.5			190; 34.8, 2.5		

*DNMT3B*, *DNA methyltransferase 3 β*; *KDM1A*, *lysine-specific histone demethylase 1A*; *MTHFR*, *methylenetetrahydrofolate reductase*; MCH, mean corpuscular hemoglobin; MCHC, mean corpuscular hemoglobin concentration; RDW, red cell distribution width; *p*-value ***, *p*-value adjusted for gestational age and days of red cell transfusions.

**Table 7 ijms-24-11817-t007:** Epistatic relationships between the genetic polymorphisms (*DNMT3B*, *KDM1A*, and *MTHFR*) in relation to erythrocyte parameters.

Epistatic Relationships	RDW (%) n; Median, Interquartile Range	*p*-Value	*p*-Value *	Erythroblasts (× 10^9^/L) n; Median, Interquartile Range	*p*-Value	*p*-Value *	MCH (pg) n; Median, Interquartile Range	*p*-Value	*p*-Value *	MCHC (pg) n; Median, Interquartile Range	*p*-Value	*p*-Value *
*DNMT3B* (Allele T) + *MTHFR* (CC)	121; 17.0, 2.8	0.062	0.543	118; 13.8, 32.9	**0.005**	0.073	121; 37.1, 3.6	**0.031**	**0.035**	124; 34.4, 2.4	**0.032**	**0.027**
Other genotypes	194; 16.6, 2.4			183; 9.0, 17.6			195; 37.6, 3.5			194; 35.0, 2.1		
*KDM1A* (AA) + *MTHFR* (CC)	23; 17.8, 5.0	**0.019**	**0.019**	21; 20.8, 13.4	**0.006**	**0.006**	23; 38.5, 4.6	0.349	0.334	24; 34.8, 2.4	0.507	0.498
Other genotypes	296; 16.7, 2.3			284; 9.6, 22.1			297; 37.4, 3.5			298; 34.9, 2.4		
*DNMT3B* (Allele T) + *KDM1A* (AA)	28; 17.7, 2.3	**0.006**	**0.009**	28; 26.0, 35.8	**0.001**	**0.019**	28; 38.0, 4.5	0.944	0.100	29; 34.9, 2.4	0.219	0.309
Other genotypes	288; 16.7, 2.3			274; 9.4, 19.4			289; 37.5, 3.5			290; 34.5, 2.4		
*DNMT3B* (Allele T) + *MTHFR* (CC) + *KDM1A* (AA)	16; 17.9, 4.8	**0.040**	**0.009**	16; 26,7, 22.7	**0.002**	**0.003**	16; 38.9, 4.9	**0.049**	0.233	17; 34.2, 2.6	0.134	0.854
Other genotypes	297; 16.7, 2.3			283; 9.7, 21.4			298; 37.4, 3.5			299; 34.9, 2.4		

*DNMT3B*, *DNA methyltransferase 3 β*; *KDM1A*, *lysine-specific histone demethylase 1A*; *MTHFR*, *methylenetetrahydrofolate reductase*; MCH, mean corpuscular hemoglobin; MCHC, mean corpuscular hemoglobin concentration; RDW, red cell distribution width; *p*-value ***, *p*-value adjusted for gestational age and days of red cell transfusions.

**Table 8 ijms-24-11817-t008:** Relationship between the different polymorphisms studied and erythrocyte parameters: (a) in the group of preterm infants who did not develop ROP; (b) in the group of preterm infants who developed ROP.

**(a) Genetic Polymorphisms in the Group without ROP**													
**Polymorphism**	**Genotype**	**RDW (%)** **n; Median, Interquartile Range**	** *p* ** **-Value**	** *p* ** **-Value ***	**Erythroblasts (× 10^9^/L)** **n; Median, Interquartile Range**	** *p* ** **-Value**	** *p* ** **-Value ***	**MCH (pg)** **n; Median, Interquartile Range**	** *p* ** **-Value**	** *p* ** **-Value ***	**MCV (fl)** **n; Median, Interquartile Range**	** *p* ** **-Value**	** *p* ** **-Value ***
*DNMT3B*	TT	42; 16.5, 1.9	0.585	0.355	40; 6.6, 17.9	0.911	0.515	42; 37.7, 4.2	0.487	0.400	42; 107.1, 9.8	0.547	0.469
	Allele C	138; 16.6, 2.4			133; 8.3, 16.2			139; 37.7, 3.2			138; 107.6, 9.7		
	CC	57; 16.6, 2.2	0.359	0.338	54; 8.0, 14.9	0.503	0.561	57; 37.5, 3.5	0.372	0.304	57; 107.7, 10.2	0.599	0.511
	Allele T	123; 16.5, 2.1			119; 8.0, 16.6			124; 37.7, 3.6			123; 107.2, 9.6		
	CT	81; 16.4, 2.4	0.695	0.894	79; 8.3, 16.6	0.473	0.311	82; 37.7, 3.3	0.807	0.776	81; 107.3, 9.6	0.984	0.966
	CC + TT	99; 16.6, 1.9			94; 7.2, 16.4			99; 37.6, 3.5			99; 107.6, 9.7		
*KDM1A*	AA	23; 16.5, 2.5	0.997	0.989	22; 7.9, 23.8	0.676	0.789	23; 38.2, 3.3	0.702	0.560	23; 105.9, 6.1	0.110	0.117
	Allele T	160; 16.6, 2.0			153; 7.6, 15.5			161; 37.7, 3.3			160; 107.9, 9.9		
	TT	77; 16.7, 1.8	0.869	0.575	74; 7.8, 16.1	0.945	0.796	77; 37.8, 3.0	0.801	0.639	76; 108.1, 9.0	0.203	0.210
	Allele A	106; 16.5, 2.4			101; 7.6, 15.1			107; 37.7, 3.6			107; 107.1, 9.4		
	TA	83; 16.4, 2.4	0.868	0.589	79; 7.6, 12.3	0.728	0.946	84; 37.6, 3.6	0.997	0.925	84; 107.5, 10.1	0.845	0.829
	AA + TT	100; 16.7, 2.0			96; 7.8, 16.7			100; 37.8, 3.0			99; 107.6, 8.5		
*MTHFR*	TT	13; 16.2, 1.4	**0.041**	**0.046**	13; 4.4, 5.7	0.068	0.385	13; 38.9, 3.5	**0.011**	**0.004**	13; 110.3, 7.0	0.060	0.082
	Allele C	167; 16.7, 2.2			160; 8.3, 16.7			168; 37.5, 3.4			167; 107.1, 9.4		
	CC	97; 16.7, 2.5	0.079	0.057	93; 10.4, 19.3	**0.022**	**0.075**	97; 37.2, 2.7	**0.002**	**0.002**	97; 106.7, 9.0	**0.033**	**0.020**
	Allele T	83; 16.5, 2.1			80; 6.3, 9.4			84; 38.4, 3.6			83; 108.3, 9.5		
	CT	70; 16.5, 2.1	0.475	0.222	67; 7.0, 9.6	0.175	0.112	71; 38.2, 3.8	0.070	0.069	70; 108.1, 10.1	0.239	0.155
	TT + CC	110; 16.5, 2.2			106; 9.2, 17.5			110; 37.4, 2.8			110; 107.1, 9.3		
**(b) Genetic polymorphisms in the group with ROP**													
**Polymorphism**	**Genotype**	**RDW (%)** **n; median, interquartile range**	** *p* ** **-value**	** *p* ** **-value *****	**Erythroblasts (× 10^9^/L)** **n; median, interquartile range**	** *p* ** **-value**	** *p* ** **-value *****	**MCH (pg)** **n; median, interquartile range**	** *p* ** **-value**	** *p* ** **-value *****	**MCV (fl)** **n; median, interquartile range**	** *p* ** **-value**	** *p* ** **-value ***
*DNMT3B*	TT	38; 16.4, 2.1	**0.010**	**0.015**	35; 14.2, 20.5	0.551	0.297	38; 36.7, 4.2	0.428	0.417	37; 108.1, 13.3	0.383	0.312
	Allele C	99; 17.6, 2.9			95; 14.9, 41.9			99; 37.4, 3.7			99; 109.2, 12.1		
	CC	29; 17.0, 2.9	0.782	0.446	28; 14.3, 18.2	0.558	0.261	29; 37.1, 1.9	0.602	0.739	29; 108.4, 8.0	0.960	0.827
	Allele T	108; 17.2, 2.8			102; 15.2, 33.6			108; 37.2, 4.5			107; 109.2, 15.0		
	CT	70; 17.6, 3.0	**0.037**	0.104	63; 18.5, 44.5	0.312	0.983	67; 37.4, 4.4	0.776	0.622	66; 109.4, 15.3	0.413	0.261
	CC + TT	67; 16.8, 2.4			67; 14.2, 19.4			70; 37.0, 3.6			70;108.2, 9.6		
*KDM1A*	AA	18; 19.1, 3.1	**0.002**	**0.006**	18; 30.9, 213.2	**0.002**	0.064	18; 38.9, 4.9	0.232	0.197	18; 112.5, 15.6	0.112	0.075
	Allele T	123; 16.9, 2.7			116; 14.0, 30.2			123; 37.1, 3.6			122; 108.1, 10.4		
	TT	58; 16.8, 2.5	0.483	0.659	56; 14.6, 42.8	0.740	0.473	58; 37.3, 5.2	0.695	0.542	58; 109.6, 13.7	0.533	0.759
	Allele A	83; 17.5, 3.1			78; 14.9, 27.8			83; 37.1, 3.8			82; 108.1, 10.3		
	TA	65; 17.0, 2.6	0.166	0.132	60; 12.9, 20.1	0.071	0.053	65; 37.4, 5.1	0.235	0.147	64; 107.5, 9.3	0.092	0.135
	AA + TT	76; 17.4, 3.0			74; 18.2, 58.8			76; 37.0, 3.4			76; 109.6, 13.7		
*MTHFR*	TT	12; 16.8, 1.8	0.864	0.767	11; 14.6, 30.2	0.636	0.298	12; 36.0, 5.2	0.124	0.111	12; 103.2, 14.6	0.073	0.095
	Allele C	128; 17.2, 3.0			122; 14.9, 33.9			128; 37.3, 4.3			127; 109.2, 13.0		
	CC	65; 17.6, 2.7	**0.008**	**0.019**	64; 17.9, 62.7	**0.031**	0.218	65;37.6, 4.7	0.472	0.461	64; 109.6, 15.0	0.079	0.059
	Allele T	75; 16.7, 2.8			69; 13.8, 22.8			75; 37.1, 3.5			75; 108.1, 10.1		
	CT	63; 16.6, 3.1	**0.011**	**0.031**	58; 13.3, 22.4	0.057	0.508	63; 37.1, 3.2	0.885	0.877	63; 108.1, 8.9	0.455	0.296
	CC + TT	77; 17.6, 2.6			133; 17.8, 42.5			77; 37.4, 4.8			76; 109.4, 15.8		

*DNMT3B*, *DNA methyltransferase 3β*; *KDM1A*, *lysine-specific histone demethylase 1A*; *MTHFR*, *methylenetetrahydrofolate reductase*; MCH, mean corpuscular hemoglobin; MCV, mean corpuscular volume; RDW, red cell distribution width; *p*-value *, *p*-value adjusted for gestational age and days of red cell transfusions.

**Table 9 ijms-24-11817-t009:** Epistatic relationships between the polymorphism of *DNMT3B*, *KDM1A*, and *MTHFR* in relation to erythrocyte parameters: (a) in the group of preterm infants who did not develop ROP; (b) in the group of preterm infants who developed ROP.

**(a) Epistatic Relationships in the Group without ROP**												
**Epistatic Relationships**	**RDW (%)** **n; Median, Interquartile Range**	** *p* ** **-Value**	** *p* ** **-Value ***	**Erythroblasts (× 10^9^/L)** **n; Median, Interquartile Range**	** *p* ** **-Value**	** *p* ** **-Value ***	**MCH (pg)** **n; Median, Interquartile Range**	** *p* ** **-Value**	** *p* ** **-Value ***	**MCV (fl)** **n; Median, Interquartile Range**	** *p* ** **-Value**	** *p* ** **-Value ***
*DNMT3B* (Allele T) + *MTHFR* (CC)	66; 16.6, 2.2	0.750	0.497	64; 9.9, 21.9	0.092	0.068	66; 37.0, 3.0	**<0.001**	**0.001**	66; 106.3, 9.2	**0.006**	**0.003**
Other genotypes	112; 16.5, 2.2			107; 7.0 12.4			113; 38.2, 3.6			112; 108.2, 9.6		
*KDM1A* (AA) + *MTHFR* (CC)	12; 16.3, 4.6	0.598	0.324	10; 10.3, 16.4	0.470	0.950	12; 38.2, 4.4	0.328	0.140	12; 105.6, 11.6	0.144	0.141
Other genotypes	167; 16.6, 2.1			162; 7.3, 16.2			168; 37.7, 3.3			167; 107.7, 9.5		
*DNMT3B* (Allele C) + *KDM1A* (AA)	13; 17.4, 2.5	0.270	0.249	13; 12.0, 34.3	0.080	0.369	13; 37.2, 3.0	0.302	0.406	13; 107.0, 8.2	0.309	0.317
Other genotypes	167; 16.6, 2.1			160; 7.3, 16.7			168; 37.7, 3.3			167; 107.7, 9.8		
*DNMT3B* (Allele C) + *MTHFR* (CC) + *KDM1A* (AA)	6; 16.6, 4.7	0.448	0.096	6; 10.3, 25.6	0.325	0.770	6; 36.8, 4.2	0.371	0.327	6; 104.1, 12.3	0.384	0.431
Other genotypes	171; 16.6, 2.2			164; 7.5, 16.7			172; 37.7, 3.3			171; 107.6, 9.7		
**(b) Epistatic relationships in the group with ROP**												
**Epistatic relationships**	**RDW (%)** **n; median, interquartile range**	** *p* ** **-value**	** *p* ** **-value *****	**Erythroblasts (× 10^9^/L)** **n; median, interquartile range**	** *p* ** **-value**	** *p* ** **-value *****	**MCH (pg)** **n; median, interquartile range**	** *p* ** **-value**	** *p* ** **-value *****	**MCV (fl)** **n; median, interquartile range**	** *p* ** **-value**	** *p* ** **-value *****
*DNMT3B* (Allele C) + *MTHFR* (CC)	55; 17.6, 2.8	0.076	0.084	54; 18.2, 8.5	**0.034**	0.265	55; 37.4, 5.0	0.642	0.664	54; 109.6, 15.2	0.103	0.100
Other genotypes	82; 16.8, 2.8			76; 13.9, 21.7			82; 37.1, 3.2			82; 108.1, 10.0		
*KDM1A* (AA) + *MTHFR* (CC)	11; 19.5, 4.8	**0.037**	**0.021**	11; 60.0, 35.9	**0.001**	**0.013**	11; 39.7, 3.0	**0.008**	**0.014**	11; 119.2, 13.9	**0.002**	**0.003**
Other genotypes	129; 17.0, 2.7			122; 14.6, 39.6			129; 37.4, 3.5			128; 108.1, 10.5		
*DNMT3B* (Allele C) + *KDM1A* (AA)	15; 19.2, 3.2	**0.011**	**0.028**	15; 26.4, 21.5	**0.006**	0.062	15; 39.0, 4.7	0.350	0.270	15; 115.6, 13.1	0.106	0.071
Other genotypes	121; 17.0, 2.7			114; 14.4, 30.8			121; 37.0, 3.6			120; 108.1, 10.8		
*DNMT3B* (Allele C) + *MTHFR* (CC) + *KDM1A* (AA)	10; 18.7, 4.5	0.066	0.063	10; 132.6, 25.1	**0.003**	0.412	10; 39.4, 3.1	**0.020**	**0.038**	10; 118.8, 14.9	**0.005**	**0.007**
Other genotypes	126; 17.0, 2.7			119; 14.6, 29.7			126; 37.0, 3.5			125; 108.1, 10.9		

*KDM1A*, *lysine-specific histone demethylase 1A*; *MTHFR*, *methylenetetrahydrofolate reductase*; *DNMT3B*, *DNA methyltransferase 3 β*; MCH, mean corpuscular hemoglobin; MCV, mean corpuscular volume; RDW, red cell distribution width; *p*-value *, *p*-value adjusted for gestational age and days of red cell transfusions.

## Data Availability

The datasets used and or analyzed during the present study are available from the corresponding author upon reasonable request.

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
