# Peer review of "Genetic Modulation of the Erythrocyte Phenotype Associated with Retinopathy of Prematurity—A Multicenter Portuguese Cohort Study"

_ijms, 2023, doi:10.3390/ijms241411817_

Round 1
Reviewer 1 Report
The manuscript "Genetic modulation of the erythrocyte phenotype associated with retinopathy of prematurity. A multicenter Portuguese cohort study" is very interesting. The Authors explained here that the polymorphisms studied may influence the susceptibility to ROP by modulating erythropoiesis and gene expression of the fetal/adult hemoglobin ratio.
Although the Authors performed all the analyses with the permission of the Ethical Committees I suggest to add the numbers of these permissions if possible.
My minor suggestions for Authors:
Table 2:
- the description of table 2 "Table 2. hematological parameters stratified by the presence or absence of ROP" should be corrected as: "Table 2. Hematological parameters stratified by the presence or absence of ROP."
Reference:
- for the reference 67: Zhang, Z.; Gao, S.; Dong, M. et al. - please add the Journal name: "Disease Markers"
Author Response
Response to Reviewer 1 Comments
Point 1. Although the Authors performed all the analyses with the permission of the Ethical Committees. I suggest to add the numbers of these permissions if possible.
Response 1: We thank your suggestion. The permission numbers of the Ethical Committees were in the “Institutional Review Board Statement” section of the manuscript. To better highlight this information, we transferred it to the “4.6. Ethics approval” section (page 23, lines 592 to 602).
Point 2. Table 2: the description of table 2 "Table 2. hematological parameters stratified by the presence or absence of ROP" should be corrected as: "Table 2. Hematological parameters stratified by the presence or absence of ROP.
Response 2: We thank your comment. The description of Table 2 has been corrected as per your suggestion (page 5, line 187).
Point 3. Reference: for the reference 67: Zhang, Z.; Gao, S.; Dong, M. et al. - please add the Journal name: "Disease Markers"
Response 3: We appreciate your comment, we agree with the observation, and we have added the Journal name: "Disease Markers" (page 27, line 827).
Reviewer 2 Report
In the article the authors address an extremely important topic of the development of retinopathy of prematurity (ROP) in the preterm newborns. The possible mechanism involved in the development of ROP is hyperoxia due to the performed care associated with the impairment of retina vascularization followed by pathological neovascularization, but it is still unclear why some preterm infants develop ROP, while others do not. The authors address the hypothesis that ROP may be associated with the genetic predisposition to the mistakes in epigenetic regulation of the genes involved in the developmental regulation, including that of the erythroid cell line genes. As a result the authors had detected association between the studied polymorphisms and development of anemia, which indirectly may also favor the development of ROP, but no connection of the selected polymorphisms with the development of ROP itself has been revealed yet.
The article is well structured and written in a clear language. All the results were correctly processed using appropriate methods of statistics.
Still there are minor corrections that can benefit the article:
1. Line 134: methylenetetrahydrofolate reductase – MTHFR is already introduced, should be used here.
2. Table 1, when indicating % of discrete variables in No ROP / ROP groups, indicates spread of the cases between groups (No ROP / ROP), while it would have been more appropriate to indicate % of cases in the group separately: e.g.: 126 male patients in No ROP group constitute 63.6 % from the total group of patients. Please, indicate % of men in No ROP group. And the same for all the variables.
3. Table 4: sum of genotypes of DNMT3B polymorphism in ROP group is equal to 99.9% at present.
4. It is better to separate tables describing genetic polymorphisms and epistatic interactions into separate tables.
5. The title of table 7 should be changed to include the description of erythrocytes parameters.
6. The introduction can be made more concise and figure 1 may be moved into discussion. At the same time, the epistatic interaction of genes may be also briefly discussed in the introduction.
7. One of the limitations of the study is its relatively low size, as associations of polymorphisms with certain conditions normally require studies of a larger scale. This may be discussed. May be associations between the studied polymorphisms and ROP development will be revealed in a larger sample.
8. Lines 495, 496, 497 references are missing
Author Response
Response to Reviewer 2 Comments
Point 1. Line 134: methylenetetrahydrofolate reductase – MTHFR is already introduced, should be used here.
Response 1: Thank you for your comment. We have changed methylenetetrahydrofolate reductase to MTHFR in line 134 (page 4).
Point 2. Table 1, when indicating % of discrete variables in No ROP / ROP groups, indicates spread of the cases between groups (No ROP / ROP), while it would have been more appropriate to indicate % of cases in the group separately: e.g.: 126 male patients in No ROP group constitute 63.6 % from the total group of patients. Please, indicate % of men in No ROP group. And the same for all the variables.
Response 2: We thank your comment. We consider it an important note and have changed the % of discrete variables in Table 1 to indicate % of cases in each group (No ROP/ ROP) separately (page 5).
Point 3. Table 4: sum of genotypes of DNMT3B polymorphism in ROP group is equal to 99.9% at present.
Response 3: Thank you for your comment. We are sorry, but we do not understand this point, as we did the calculations again for the various genotype combinations of the DNMT3B polymorphism in the ROP group and obtained 100.0%. If you continue to see this error, could you please indicate in which genotypes of the DNMT3B polymorphism in the ROP group did you get this result? Thank you.
Point 4. It is better to separate tables describing genetic polymorphisms and epistatic interactions into separate tables.
Response 4: We appreciate your comment. We consider this an important observation and have separated the tables describing genetic polymorphisms and epistatic interactions into different tables. For this purpose, table 4 has been separated into two tables, table 4 and a new table 5; the initial table 5 has been separated into two tables, a new table 6 and a new table 7. Due to these changes, the initial table 6 became Table 8 and the initial table 7 became Table 9. As a result of these changes to the tables, the text of lines 203 to 205, which referred to table 4 (b) (current table 5), has changed to lines 209 to 211, which are located between the current tables 4 and 5.
Point 5. The title of table 7 should be changed to include the description of erythrocytes parameters.
Response 5: Thank you for your comment. We agreed with the observation, and we changed the title of table 7 to include the description of erythrocyte parameters (page 15, line 259). Please note that in this new version of the manuscript the initial table 7 is now table 9.
Point 6. The introduction can be made more concise and figure 1 may be moved into discussion. At the same time, the epistatic interaction of genes may be also briefly discussed in the introduction.
Response 6: We appreciate your comment, we consider it an important note. To make the introduction more concise, we removed a sentence whose content we thought was less relevant (page 2: lines 85 and 86) and changed some sentences from the introduction to the discussion (lines 98 to 101 on page 3 changed to lines 314 to 317 on pages 17 and 18; lines 122 to 124 on page 3 changed to lines 318 to 320 on page 18). We moved figure 1 into discussion (page 18). We have also added a brief discussion of the epistatic interaction of genes to the introduction (page 4, lines 154 to 159).
Point 7. One of the limitations of the study is its relatively low size, as associations of polymorphisms with certain conditions normally require studies of a larger scale. This may be discussed. May be associations between the studied polymorphisms and ROP development will be revealed in a larger sample.
Response 7: We thank your comment. We agree with this important observation and have added this limitation to the discussion (page 21, lines 499 to 502).
Point 8. Lines 495, 496, 497 references are missing.
Response 8: Thank you for your comment. We have added the references to the manuscript, they are references 68 to 70 (page 22, lines 541 to 543).